# Residual Cystine Transport Activity for Specific Infantile and Juvenile *CTNS* Mutations in a PTEC-Based Addback Model

**DOI:** 10.3390/cells13070646

**Published:** 2024-04-06

**Authors:** Louise Medaer, Dries David, Maxime Smits, Elena Levtchenko, Maurilio Sampaolesi, Rik Gijsbers

**Affiliations:** 1Laboratory of Molecular Virology and Gene Therapy, Department of Pharmacological and Pharmaceutical Sciences, Faculty of Medicine, KU Leuven, 3000 Leuven, Belgium; louise.medaer@kuleuven.be (L.M.); maxime.smits@kuleuven.be (M.S.); 2Leuven Viral Vector Core, Faculty of Medicine, KU Leuven, 3000 Leuven, Belgium; 3Department of Paediatric Nephrology & Development and Regeneration, University Hospitals Leuven & KU Leuven, 3000 Leuven, Belgium; e.n.levtchenko@amsterdamumc.nl; 4Department of Paediatric Nephrology, Amsterdam University Medical Centre, 1081 Amsterdam, The Netherlands; 5Translational Cardiology Laboratory, Department of Development and Regeneration, Stem Cell Institute, Faculty of Medicine, KU Leuven, 3000 Leuven, Belgium; maurilio.sampaolesi@kuleuven.be

**Keywords:** cystinosis, kidney disease, CTNS^mutants^, gene therapy, viral vectors

## Abstract

Cystinosis is a rare, autosomal recessive, lysosomal storage disease caused by mutations in the gene *CTNS*, leading to cystine accumulation in the lysosomes. While cysteamine lowers the cystine levels, it does not cure the disease, suggesting that CTNS exerts additional functions besides cystine transport. This study investigated the impact of infantile and juvenile *CTNS* mutations with discrepant genotype/phenotype correlations on CTNS expression, and subcellular localisation and function in clinically relevant cystinosis cell models to better understand the link between genotype and CTNS function. Using CTNS-depleted proximal tubule epithelial cells and patient-derived fibroblasts, we expressed a selection of CTNS^mutants^ under various promoters. *EF1a*-driven expression led to substantial overexpression, resulting in CTNS protein levels that localised to the lysosomal compartment. All CTNS^mutants^ tested also reversed cystine accumulation, indicating that CTNS^mutants^ still exert transport activity, possibly due to the overexpression conditions. Surprisingly, even CTNS^mutants^ expression driven by the less potent *CTNS* and *EFS* promoters reversed the cystine accumulation, contrary to the CTNS^G339R^ missense mutant. Taken together, our findings shed new light on *CTNS* mutations, highlighting the need for robust assessment methodologies in clinically relevant cellular models and thus paving the way for better stratification of cystinosis patients, and advocating for the development of more personalized therapy.

## 1. Introduction

Cystinosis is a monogenic, autosomal recessive, lysosomal storage disease caused by biallelic mutation in the *CTNS* gene (17p13.2) [1,2,3]. This gene encodes cystinosin (CTNS, 367 AA), a H^+^-cystine symporter transporting cystine to the cytosol (Figure 1) [4,5]. It includes seven transmembrane (TM) domains with seven predicted N-glycosylation sites at the N-terminus, and lysosomal targeting motifs located at the C-terminus (GYDQL) and the PQ-loop in the fifth inter-transmembrane (IT) region (YFPQA). *CTNS* mutations are shown to cause a defect in the CTNS cystine transport activity and thus lead to an accumulation of cystine in the lysosomes of all body cells and tissues, making cystinosis a systemic disease with the kidney and the eyes being the first organs to be affected [6,7,8,9]. In the clinic, three phenotypes are distinguished: nephropathic infantile, nephropathic juvenile, and non-nephropathic ocular cystinosis [10]. Patients with nephropathic infantile cystinosis (OMIM 219800) appear normal at birth but early clinical manifestations (around 6–9 months of age) include a failure to thrive and rickets because of the generalized dysfunction of kidney proximal tubule cells (PTECs), also called renal Fanconi syndrome. If left untreated, infantile cystinosis leads to end-stage kidney disease (ESKD) by the age of 10 [11,12,13]. In the case of nephropathic juvenile cystinosis (OMIM 219900), patients are diagnosed later in childhood or during adolescence, with milder forms of the renal Fanconi syndrome or isolated proteinuria with a slower rate of progression towards ESKD [13,14,15]. The non-nephropathic ocular phenotype (OMIM 21975), diagnosed in adults, is mild and characterized photophobia due to cystine accumulation in the cornea and conjunctiva of the eyes without systemic organ damage [10,12,16]. For treating cystinosis, the beneficial role of cysteamine therapy has been well described for nearly four decades. Although it does not reverse proximal tubulopathy, it considerably delays progression towards kidney failure and postpones non-renal complications [17,18].

To date, over 165 mutations have been reported for cystinosis, which include 69 missense and nonsense mutations, 23 splicing mutations, 52 deletions, 15 insertions, four indels, and two promoter region mutations [19,20]. The most common pathogenic mutation, representing approximately 50% of mutant alleles in the Caucasian population, is a 57-kb deletion encompassing the *CTNS* promoter together with the first nine exons and part of exon 10 together with the gene *CARKL* and the first two non-coding exons of *TRPV1* upstream of *CTNS* [14,21,22,23]. The effect of *CTNS* missense mutations and small indels on protein expression, subcellular distribution, and function is only poorly understood, and has been primarily studied in overexpression conditions (transient transfection) with CTNS proteins carrying an outspoken eGFP tag (27 kDa), and mostly in non-human cell lines [24,25,26,27]. Moreover, to assess transport activity, CTNS was reengineered by removing the C-terminal lysosomal targeting motif GYDQL to redirect CTNS and CTNS mutant proteins to the plasma membrane (PM) [26]. Recently, the crystal structures of *Homo sapiens* and *Arabidopsis thaliana* CTNS were solved in lumen-open, cytosol-open, and cystine-bound states by crystallography and cryo-EM, revealing the cystine recognition mechanism and key conformational states of the proton-coupled transport cycle [27,28]. These studies, together with Alphafold and AlphaMissense provided us with a framework for a better understanding of the genotype–phenotype interplay, allowing us to explore the impact of missense mutations causing cystinosis on CTNS’ function [29,30].

In this study, we set out to determine the effect of specific point mutations and expression levels on CTNS activity in a clinically relevant kidney-derived proximal tubule cell model using stable lentiviral vector (LV)-mediated expression. We studied a subset of CTNS missense mutations and a small deletion that causes either infantile or juvenile phenotypes in patients living with cystinosis with discrepant genotype/phenotype correlations, and assessed expression, subcellular location, and cystine accumulation at different expression levels.

## 2. Materials and Methods

### 2.1. Cell Culture

The conditionally immortalized proximal tubulus epithelial cells (ciPTECs) cell lines used in this study are the following: a healthy control ciPTEC 14.4 cell line (referred to as *CTNS^+/+^*) and a CTNS-depleted ciPTEC cell line (referred to as *CTNS*^−/−^) derived from ciPTEC 14.4. ciPTECs 14.4 were generated by the isolation of PT cells obtained from urine from healthy volunteers and transfected with a temperature-sensitive mutant U19tsA58 of SV40 large T antigen (SV40T) and the essential catalytic subunit of human telomerase as described earlier [31]. *CTNS*^−/−^ ciPTECs were described earlier (kind gift from Dr. Janssen M. and Prof. Masereeuw R., Utrecht University, the Netherlands) [32]. *CTNS*^−/−^ ciPTEC is an isogenic clone derived from ciPTEC 14.4 and generated using CRISPR-Cas9, which harbors a 13-base pair (bp) and 85 bp deletion in exon 4 of the *CTNS* gene. The cells were cultured in Dulbecco’s modified Eagle DMEM F-12 (L0093-500; Biowest, Leuven, VWR Belgium) supplemented with 5 mL/500 mL insulin–transferrin–selenium (I-1884; Sigma, Overijse, Belgium), 36 ng/mL hydrocortisone (Sigma, H0135), 10 ng/mL EGF (E9644; Sigma, Overijse, Belgium), 40 pg/mL tri-iodothryonine (T5516, Sigma, Overijse, Belgium), 10% fetal bovine serum (*v*/*v*; DE17-602E; Biowest, Leuven, VWR Belgium), and 1.1% Pen/Strep (DE17-602E; Westburg, Leusden, The Netherlands) [32]. After transduction, the cells were selected with puromycin (1 μg/mL; ant-pr-1; Invivogen, Toulouse, France). ciPTECs were grown at 33 °C and 5% (*v*/*v*) CO_2_ for proliferation up to 90% confluency and matured into differentiated epithelial cells by culturing at 37 °C for 7 days [31].

The Fcys (*CTNS*^−/−^; Fcys32) fibroblast cell line, isolated from skin samples of a cystinosis patient, was developed as described earlier [33,34]. The genotype consists of a homozygous 57 kb deletion of the *CTNS* gene. The FCo (*CTNS*^+/+^; FCo5) control fibroblast cell line was derived from a healthy volunteer [33,34]. The cells were cultured in DMEM/F-12 medium (Invitrogen, Merelbeke, Belgium), supplemented with 10% fetal bovine serum (*v*/*v*; DE17-602E; Biowest, Leuven, VWR Belgium), L-glutamine (4 mM, Invitrogen, Merelbeke, Belgium), penicillin (100 U/mL; Invitrogen, Merelbeke, Belgium), and streptomycin (100 µg/mL; Invitrogen, Merelbeke, Belgium). The fibroblasts were grown at 37 °C and 5% (*v*/*v*) CO_2_ for proliferation.

### 2.2. Generation of LV Transfer Plasmids for CTNS^WT^ or CTNS^mutants^

The human *CTNS^WT^*, *CTNS^mutants^*, *eGFP*, and *dATP13A2* cDNA were cloned in the self-inactivating lentiviral (SIN-LV) backbone plasmid pCH-promoter-X-IRES-PuroR-WPRE (Didier Trono) using Gblocks at the BcuI and Bsp119I4 restriction sites. The cDNA constructs encoding eGFP and dATP13A2 (D508N), a catalytically dead version of ATP13A2, a lysosomal transmembrane protein, were used as transduction controls and to control for overexpression. The vector backbone contained one of the following promoters: *CMV*-, *EF1a*, *EFS* (EF1a short), or *CTNS* promoter [35,36]. The *CTNS^WT/mutant^* cDNA was tagged with a triple hemagglutinin tag (3HA tag) at its C-terminus.

### 2.3. Production of LV Vectors and Generation of Stable Cell Lines Expressing CTNS^WT^ or CTNS^mutants^ Driven by Different Promoters

LVs were produced as previously reported [37]. Functional validation of the LV_CTNS^WT^-3HA constructs was reported in Veys et al. [38]. To ensure a single integrated viral vector copy per cell, viral vector transduction was conducted employing a limiting dilution series. Cells were seeded in a 96-well plate to reach the ideal confluency at 10,000 cells, grown overnight, and transduced with the respective LV vector preparations. A total of 72 h later, the medium was replaced with puromycin-containing (1 µg/mL, ant-pr-1; Invivogen, Toulouse, France) medium to select transduced cells. To ensure a single integrated viral vector copy, we selected the highest dilution that still resulted in surviving cells upon puromycin selection (<20% transduced cells, MOI < 0.5) [39].

### 2.4. Determination of Integrated Copies

gDNA was extracted using the GenElute^TM^ Mammalian Genomic DNA miniprep Kit (Sigma, Overijse, Belgium) according to the manufacturer’s protocol. A total of 25 ng/µL gDNA was used for qPCR to determine integrated copies based on the woodchuck hepatitis virus post-transcriptional regulatory element (WPRE). As a control, a single integrated copy control (1ICC) was made generated by transducing HEK293T cells in a limiting dilution series with an LV expressing pEF1a-CTNS^WT^-eGFP-IRES-PuroR. Cells were monitored by flow cytometry analysis, and the condition where the Mean Fluorescence Intensity (MFI) did not drop when the vector was diluted, but the % eGFP positive cells did, was considered as the 1ICC condition. Following puromycin selection, the highest dilution that survived the selection was chosen. RT-qPCR was performed using 25 ng/µL gDNA, dsDNA-intercalating agent LightCycler^®^ 480 SYBR Green I (Roche Life Science, Brussels, Belgium), and 10 µM primers (see Appendix A). RT-qPCR was performed on the CFX Opus 96 Real-Time PCR instrument (Bio-Rad, Temse, Belgium) and data were retrieved and analysed using the CFX maestro 2.2 software. Amplification was performed for 50 cycles of 10 s at 95 °C and 30 s at 60 °C. The fold change was calculated as fold change = 2^−∆∆Ct^.

### 2.5. Quantification of CTNS-3HA mRNA Expression Levels

Total mRNA was extracted using the AurumTM Total RNA Mini Kit (Bio-rad, Temse, Belgium) following the manufacturers’ instructions. cDNA was synthesized from the extracted mRNA samples using the High-Capacity cDNA Reverse Transcription Kit (Applied Biosystems, Merelbeke, Belgium). RT-qPCR was performed using 5 ng/µL cDNA, dsDNA-intercalating agent LightCycler^®^ 480 SYBR Green I (Roche Life Science, Brussels, Belgium), and 10 µM primers. Primers were designed to land in exonic sequences, spanning exon 10 and 11, allowing for the assessment of endogenous mRNA- and LV-expressed *CTNS* mRNA (Appendix A). RT-qPCR was performed on the CFX Opus 96 Real-Time PCR instrument (Bio-Rad, Temse, Belgium) and data were retrieved and analysed using the CFX maestro 2.2 software. Amplification was performed for 50 cycles of 10 s at 95 °C and 30 s at 60 °C. The fold change was calculated as fold change = 2^−∆∆Ct^.

### 2.6. Metabolite and Cystine Measurements

The ciPTECs were seeded at 55,000 cells/cm^2^ in a 6-well plate and allowed to differentiate at 37 °C for 7 days. Samples were prepared by removing the medium and washing the cells with a 0.9% NaCl solution. The washing solution was removed and the extraction buffer was added. The extraction buffer with cystine internal standard is prepared as follows. A 20 mM ^15^N_2_-Cystine stock standard was prepared by dissolving 4.8 mg ^15^N_2_-cystine (Cambridge Isotope Laboratories NLM-3818; Apeldoorn, The Netherlands) in 1 mL of a 30/70 volume ratio mix of respectively, 2M HCl (fuming 37%, 1.00317.1000 Merck, Hoeilaart, Belgium) in milliQ water and methanol (85800.320; VWR, Leuven, Belgium). This solution was then diluted to insert the final concentration in a solution of 80/20 methanol/milliQ water with 0.1 v% formic acid (85048.001; VWR, Leuven, Belgium). Using a cell scrape, the extract was transferred into an Eppendorf tube. Proteins were pelleted by centrifugation for 15 min at 20,000× *g* at 4 °C and used to normalization, determined by a Pierce^TM^ BCA protein assay (Thermo Scientific, Dilbeek, Belgium). The supernatant was transferred to a new Eppendorf to perform mass spectrometry. Mass spectrometry measurements were performed using a Vanquish LC System (Thermo Scientific, Dilbeek, Belgium) coupled via heated electrospray ionization to a Q Exactive Orbitrap Focus mass spectrometer (Thermo Scientific, Dilbeek, Belgium). A 10 μL sample was taken from an MS vial and injected onto a Poroshell 120 HILIC-Z PEEK Column (Agilent InfinityLab, Zaventem, Belgium). A linear gradient was carried out starting with 90% solvent A (acetonitrile with 5 µM medronic acid) and 10% solvent B (10 mM NH_4_-formate in milli-Q water, pH 3.8). From 2 to 12 min the gradient changed to 60% B. The gradient was kept on 60% B for 3 min and followed by a decrease to 10% B. The chromatography was stopped at 25 min. The flow was kept constant at 0.25 mL/min and the column was kept at 25 °C throughout the analysis. The mass spectrometer operated in full scan (range [70.0000–1050.0000]) and positive mode using a spray voltage of 3 kV, capillary temperature of 320 °C, sheath gas at 45, auxiliary gas at 10, and the latter heated to 260 °C. The AGC target was set at 3.0E+006 using a resolution of 70,000. Data collection was performed using the Xcalibur software 4.2.47 (Thermo Scientific, Dilbeek, Belgium). The data analyses were performed by integrating the peak areas (El-Maven–Polly–Elucidata), and cystine was quantified using the known concentration of ^15^N_2_-cystine spiked in the extraction buffer. Metabolites from glycolysis, the Krebs cycle, amino acids, nucleotides, energy charge, and redox molecules were measured in addition to cystine. The data are depicted as abundancies (log scale) or µM cystine normalized for total protein content. MetaboAnalyst 6.0 software was used to generate volcano plots with a fold change (FC) threshold of 4.0 and *p*-value threshold of 0.05 (two-sample unpaired *t*-test).

### 2.7. Protein Analysis by PAGE and Western Blot

Cell pellets were homogenized in 1% Sodium Dodecyl Sulfate (SDS, Sigma) together with protease inhibitors (Merck, Hoeilaart, Belgium) and subsequently heated for 5 min at 98 °C. After sonication, the samples were again heated for 5 min at 98 °C. The total protein concentration was determined using the Pierce^TM^ BCA Protein Assay Kit (Thermo Scientific, Dilbeek, Belgium) following the manufacturers’ instructions. A total of 10–20 µg of protein was mixed with SDS loading dye (6x) containing β-mercapto-ethanol (10%; VWR, Leuven, Belgium) and subsequently loaded on a 4–15% tris-glycine gel (4–15% Criterion™ TGX™ Precast Midi Protein Gel). The proteins were transferred using a Trans-Blot Turbo Transfer System (Biorad, Temse, Belgium). After transfer, the PVDF membrane was blocked using 5% milk in 0.1% PBS-tritonX-100 solution (215682500; Acros Organics, Geel, Belgium) and subsequently incubated with primary antibodies overnight (ON) at 4 °C (the specification of all antibodies is described in Appendix A). Next, the membrane was incubated with a secondary anti-species antibody conjugated with horse radish peroxidase (HRP; Dako Agilent, Leuven, Belgium) in 5% milk in 0.1% PBS-triton solution. The proteins were visualized following incubation with the Clarity™ Western ECL Substrate (Biorad, Temse, Leuven). Images were acquired using the LAS-3000 Imaging system (Fuji, Zaventem, Belgium) or Amersham ImageQuant 800 Fluor (GE Healthcare, Diegem, Belgium) and subsequently quantified with ImageQuant^TM^ TL software V8.2.

To remove N-linked oligosaccharides from glycoproteins, PNGase F treatment (New England Biolabs; Leiden, The Netherlands) was performed following the manufacturers’ instructions. Subsequently, samples were processed as indicated before.

### 2.8. Immunocytochemistry Staining

ciPTECs were seeded in non-detachable chambers (Ibidi, Beloeil, Belgium) and, after 72 h, fixed with 4% paraformaldehyde (PFA; Sigma, Overijse, Belgium) for 20 min at RT. To permeabilize, the ciPTECs were incubated with 0.1% triton X-100 (Acros Organics, Geel, Belgium) for 30 min. The cells were washed with PBS and subsequently blocked for 30 min with 1% bovine serum albumin (BSA; Sigma). After washing with PBS, the cells were incubated ON at 4 °C with primary antibodies diluted in PBS with 0.1% BSA. The next day, the cells were washed with PBS and incubated for 30 min with secondary antibodies (Thermo Scientific, Dilbeek, Belgium) diluted in PBS with 0.1% BSA and DAPI (1/2000, Sigma) diluted in Mowiol mounting medium (Sigma, Overijse, Belgium).

The samples were visualized with a laser scanning confocal microscope (Zeiss LSM 780, Cell Imaging Core (KU Leuven)) in combination with a Plan-Apochromat 63x/1.4 Oil DIC M27 objective (Zeiss, Brussel, Belgium), and the following lasers: 488 nm, 561 nm, 633 nm, and 405 nm (DAPI).

### 2.9. AlphaMissense Pathogenicity and CADD Scores

We used AlphaMissense via the web resource https://alphamissense.hegelab.org (accessed on 24 January 2024), providing us with the pathogenicity scores and PDBe-mol structure viewer containing the predicted structure of CTNS from AlphaFold using CTNS in the search interface: CTNS_HUMAN, O60931, ENST00000046640.7 [29,40]. We confirmed the subset of mutations in AlphaMissense, which uses pathogenicity scores and classifies it as either likely benign, likely pathogenic, or uncertain based on the structural context of variants (AlphaFold), their evolutionary conservation, and protein language modelling (Table 1) [29,40,41]. The AlphaMissense pathogenicity score of the variants is given as the log-likelihood difference of a residue relative to the reference residue at that position. The Combined Annotation Dependent Depletion (CADD) algorithm measures the deleteriousness of genetic variants. Pre-computed CADD scores for the respective CTNS missense mutants were retrieved from the PopViz webserver (https://hgidsoft.rockefeller.edu/PopViz/, accessed on 24 January 2024) [42].

### 2.10. Statistical Analysis

Data are expressed as the mean ± standard deviation with individual data points shown in each group (replicates of multiple independent experiments). GraphPad 8.0.2 was used to plot all graphs and perform statistical analysis.

## 3. Results

### 3.1. Description of the Selected CTNS Mutations

Until now, most in vitro models have studied infantile cystinosis in the context of the 57-kb deletion, resulting in complete loss of the CTNS protein. However, a subset of patients suffers from cystinosis and carry amino acid substitutions [20]. In a recent cohort study, it was shown that 23% of the patients were heterozygous for 57-kb deletion, and 45% had other pathogenic mutations in the *CTNS* gene [23]. In this study, we set out to determine the impact of *CTNS* missense mutations and a small deletion on CTNS transport activity in a clinically relevant kidney-derived proximal tubule epithelial cell model. We selected a subset of CTNS mutations that were located over the whole CTNS protein, and correlated with either infantile or juvenile phenotypes in cystinosis patients: Del.67–73, W182R, K280R, N288K, S298N, and N323K (Figure 1). These mutants were selected because of the discrepancy between the clinical phenotype and the in vitro cystine transport activity reported by Kalatzis et al. [24]. We collected all published information for these mutants in Table 1. Additionally, we determined AlphaMissense pathogenicity and CADD scores for the respective CTNS missense mutants [20,24,29]. W182R, K280R, N288K, S298N, and N323K are predicted to be likely_pathogenic with an AlphaMissense pathogenicity score between 0.5957–0.9942, and CADD scores ranging between 22.6–35.

### 3.2. Both CTNS^WT^ and CTNS^mutants^ Restore Cystine Content in Cystinosis Cell Models upon Overexpression

For each of the selected mutants, HIV-based lentiviral vectors (LV) were constructed, driving the expression of the respective *CTNS^mutant^* cDNAs including a C-terminal triple hemagglutinin tag (3HA) from the ubiquitous human *EF1a* promoter (LV_pEF1a-CTNS^mutant^; Figure 2A)*. CTNS* CRISPR-ed (CTNS depleted) conditionally immortalized human proximal tubule epithelial cells (referred to as *CTNS^−/−^* ciPTECs) were transduced with the respective LVs at low multiplicity of infection (MOI) to ensure single integrated copies, and selected [32]. As a reference, wild-type (WT) *CTNS* cDNA was taken along (LV_pEF1a-CTNS^WT^). To control for the transduction of CTNS, first a stable cytosolic eGFP overexpression cell line was taken along (LV_ctrl). *CTNS* mRNA levels in CTNS^WT^- and CTNS^mutant^-transduced *CTNS*^−/−^ ciPTECs were >2 logs higher compared to the endogenous *CTNS* expression levels in reference *CTNS*^+/+^ ciPTECs (Figure 2B). Furthermore, the protein levels were examined by Western blot analysis (Figure 2C). Because of the heavily glycosylated N-terminus, CTNS^WT^ and the respective CTNS^mutants^ were observed as a diffuse band at ~70 kDa [45]. Removal of the N-linked oligosaccharides by PNGase F treatment shifted the respective proteins to ~41 kDa. For the deletion mutant, CTNS^del.67-73^, PNGaseF treatment did not affect the migration in the Western analysis, in line with previous reports [45]. Next to normal protein expression, all missense CTNS mutants were located on the lysosomes, as shown by the subcellular co-localisation of CTNS^WT^ and CTNS^mutants^ with lysosomal-associated membrane protein 1 (LAMP1/LA1) (Figure 2D). After validation of protein expression and correct subcellular localisation following stable expression of CTNS^WT^ and CTNS^mutants^, we performed metabolomic analysis and evaluated the cystine levels. First, we evaluated parental *CTNS*^+/+^ and the CRISPRed *CTNS*^−/−^ cells. Here, no significant alterations were detected in metabolites from glycolysis, the Krebs cycle, energy charges, amino acids, or nucleotides except for cystine, cysteine, and GSSG (Figure 2E and Appendix A; *p* < 0.05, unpaired *t*-test). CTNS^WT^ and CTNS^mutant^ addition resulted in a significant reduction in the cystine levels compared to transduction control (compared to LV_ctrl; *p* < 0.0001, one-way Anova, Sidak’s multiple comparison test), resulting in cystine levels like parental non-cystinosis cells (*CTNS^+/+^*) (Figure 2E). Additionally, the elevated cysteine and GSSG/(GSH + GSSG) ratio in *CTNS*^−/−^ ciPTECs, known to be altered in cystinosis, showed normalization upon overexpression of any of the CTNS^mutant^ proteins (Appendix A) [64]. Similarly, all CTNS^mutants^-complemented cystinosis patient-derived fibroblasts showed restored cystine accumulation to CTNS^WT^ levels (Appendix A; compared to LV_ctrl; *p* < 0.0001, one-way Anova, Sidak’s multiple comparison test). Taken together, we showed that CTNS^mutant^ proteins expressed well and located to the lysosome as CTNS^WT^. Moreover, cystine measurements illustrated that a stable introduction of the CTNS^mutant^ proteins in *CTNS*^−/−^ ciPTECs and patient-derived fibroblasts allowed cystine accumulation to revert to levels observed in CTNS^WT^-complemented ciPTECs and fibroblasts.

### 3.3. Physiologically More Relevant CTNS^WT^ Expression Levels Rescue Cystine Accumulation

We hypothesized that the rescue of cystine accumulation in the abovementioned cell models could be explained by the supraphysiological expression levels of *CTNS^mutant^* mRNA and CTNS^mutant^ protein. To explore more physiological expression levels, we complemented *CTNS*^−/−^ ciPTECs with *CTNS^WT^* cDNA carrying a less potent *EFS* promoter (EF1a-short; p*EFS*) and the *CTNS* promoter (p*CTNS*) to drive transcription, and examined cystine accumulation (Figure 3A) [35,36]. *CTNS* and *EFS* promoter functionality in ciPTECs was confirmed by eGFP expression analysis for LV_eGFP control constructs (LV_pCTNS-eGFP, LV_pEFS-eGFP), demonstrating a 7- and 1.3-fold lower MFI (based on one integrated copy; ~30% eGFP^+^ cells) compared to p*EF1a*-driven expression (Appendix A). Instead of using eGFP, we from here on, employed control expression constructs of a dead ATP13A2 protein (dATP13A2), a lysosomal transmembrane protein [65]. Next, we assessed *CTNS*^−/−^ ciPTEC transduced with the respective promoter–CTNS^WT^ constructs. Integration of the transgene construct was confirmed by measuring integrated copies (Appendix A). Expression levels of *CTNS^WT^* driven by the *EF1a* promoter resulted in a 160-fold higher expression than endogenous expression in *CTNS^+/+^*, while the *EFS* and *CTNS* promoter showed 26- and 18-fold higher expression levels, respectively (Figure 3B). CTNS^WT^-3HA protein expression was shown by Western blot analysis (Figure 3C). Quantification of the Western blot signals showed a 4- and 33-fold lower protein expression compared to *EF1a*-driven expression, when *CTNS^WT^-3HA* is driven by the *EFS* or *CTNS* promoter, respectively (Appendix A). For all three promoters, we confirmed lysosomal CTNS^WT^ expression by immunocytochemistry staining (Appendix A). Next, we performed a mass spectrometry analysis to assess the levels of cystine accumulation and demonstrated that cystine levels decreased to *CTNS^+/+^* (and LV_pEF1a-CTNS^WT^) levels in *CTNS*^−/−^ ciPTECs after *CTNS^WT^* cDNA addition driven by the *EFS* and *CTNS* promoter compared to transduction control (Figure 3D; compared to LV_ctrl; *p* < 0.0001, one-way Anova, Sidak’s multiple comparison test). Additionally, the cysteine levels and redox state normalized to parental non-cystinosis *CTNS^+/+^* levels, confirming that lower and physiologically more relevant CTNS expression levels were sufficient to reverse the cystinosis phenotype (Appendix A). This prompted us to further evaluate the effect of CTNS^mutants^ at lower expression levels.

### 3.4. CTNS^mutant^ Expression Driven by EFS- and CTNS Promoter Still Reverts Cystine Levels

We transduced *CTNS*^−/−^ ciPTECs with LV vectors encompassing *CTNS^mutants^* cDNA driven by the *EFS* or the *CTNS* promoter. Integration of the transgene construct was confirmed by measuring integrated copies (Appendix A). For both promoters, CTNS^mutant^ protein expression was confirmed by Western blot analysis, showing substantially lower proteins levels for *CTNS* promoter-driven constructs (with some of the CTNS^mutants^ barely detectable), as expected with CTNS^del67-73^ and CTNS^W182R^ showing lower protein levels in both conditions (Figure 4B). The enhanced contrast blot shows CTNS^del67-73^ at 25 kDa and visualizes the low expression levels of *CTNS* promoter-driven constructs (Appendix A). Scanning of the Western blot signals showed on average a 14-fold lower protein expression level when *CTNS^mutants^* cDNA was driven by the *CTNS* promoter (Appendix A). Still, these physiologically more relevant expression levels of CTNS^mutants^ reduced the cystine content in *CTNS*^−/−^ ciPTECs significantly compared to the transduction control (*p* < 0.0001 compared to LV_ctrl, one-way Anova, Sidak’s multiple comparison test), reaching *CTNS^+/+^* levels for most of the conditions, even though the protein levels of CTNS^mutants^ were substantially lower for the *CTNS* promoter-driven constructs (Figure 4C). Interestingly, two conditions, p*CTNS*-CTNS^del67-73^ and p*CTNS*-CTNS^N288K^, resulted in a significant drop in cystine accumulation compared to the transduction control (*p* < 0.0001, one-way Anova, Sidak’s multiple comparison test), but were still significantly higher than pCTNS-CTNS^WT^ (*p* < 0.0001 compared to LV_CTNS^WT^, one-way Anova, Sidak’s multiple comparison test). Even though the results obtained were exciting, the fact that cystine accumulation in most cases was completely rescued in our cell model, even when using expression levels that are close to physiological levels, prompted us to demonstrate that this is not the case for every single *CTNS* missense mutation. Therefore, we additionally selected another missense CTNS mutation, G339R, that is associated with the infantile form of cystinosis, and is predicted to be pathogenic based on the AlphaMissense pathogenicity score and the CADD score (0.9803 and 29, respectively; Table 1). We complemented *CTNS*^−/−^ ciPTECs with LV-encoding *CTNS^G339R^* cDNA driven by the *EFS* promoter and selected the transduced cells. Integration of the *CTNS^G339R^* transgene construct was confirmed by measuring integrated copies (Appendix A). CTNS^WT^ and CTNS^K280R^ were taken along as controls. Protein expression was corroborated by Western blot analysis and immunohistochemistry, demonstrating that CTNS^G339R^, in line with the other CTNS point mutations assessed, expressed well and localised to the lysosomes (Figure 5A,B). Contrary to CTNS^WT^ and CTNS^K280R^, CTNS^G339R^ did not restore the cystine accumulation, underscoring that not all *CTNS* missense mutations behave the same (Figure 5C). Similarly, cysteine accumulation was also not restored (Appendix A).

## 4. Discussion

Hitherto, the study of cystinosis has largely focused on the most common Caucasian mutation resulting in complete loss of the CTNS protein due to a 57 kb deletion [14,21,22]. Still, close to 50% of patients living with cystinosis carry point mutations and smaller indels in the CTNS protein and the effect of these mutations on the protein and its function is only poorly studied. We used a clinically relevant CTNS-depleted kidney cell model and patient-derived *CTNS*^−/−^ fibroblasts to stably express specific CTNS mutants from different promoters (Table 1) [32]. The mutations selected were associated with infantile and juvenile nephropathic cystinosis and were selected based on the discrepancies between in vitro cystine transport activity observed by Kalatzis et al. and the clinical phenotype [24]. Their pathogenicity scores based on AlphaMissense and CADD score indicated that all mutations tested were likely to be pathogenic, in line with the phenotype of the patients carrying the mutation (Table 1). When introducing the *CTNS^mutants^* driven by the *EF1a* promoter in CTNS-depleted cells (patient-derived fibroblasts or CRISPRed *CTNS*^−/−^ PTECs) using LVs, all proteins expressed well, located to the lysosome, and normalized cystine to WT levels (*CTNS^+/+^*) (Figure 2D). *EFS* promoter- and *CTNS* promoter-driven CTNS^WT^ protein expression levels were significantly lower compared to the *EF1a* promoter-driven constructs (4- to 33-fold, respectively, Figure 3B,C) and rescued cystine accumulation. Still, the mRNA expression levels were 18–26-fold higher than the endogenous *CTNS* mRNA in *CTNS^+/+^* ciPTECs, suggesting that the endogenous *CTNS* promoter together with additional regulatory elements limits expression [35,66]. A possible explanation may be the fact that our constructs only include the open reading frame of *CTNS*, lacking the UTR and intronic sequences, which may contribute to the higher mRNA levels. Moreover, the LV constructs encompass an RNA stabilizing WPRE sequence, and the integrated lentiviral vector copy number, which also contribute to the overall higher *CTNS* expression levels [67]. We detected more integrated copies when CTNS^mutants^ and the puromycin selection cassette was driven by the weaker *CTNS* promoter, resulting in higher expression levels (Appendix A). Intriguingly, also when using the less potent *EFS* and *CTNS* promoters, all CTNS^mutants^ restored the cystine accumulation to WT levels when introduced in *CTNS*^−/−^ ciPTECs, suggesting that this subset of CTNS^mutants^ were still functional, even at near physiological expression levels. CTNS^del.67-73^ and CTNS^N288K^ showed significant but incomplete rescue of cystine accumulation, in contrast with the other mutants when driven by the *CTNS* promoter (Figure 4C). Both were reported to degrade faster compared to CTNS^WT^ [45,49]. In addition, CTNS^Del.67-73^ was shown to be unable to exit the ER. CTNS^N288K^ is located at the 5th inter-transmembrane loop, which contains the PQ-motif, important for H^+^ and cystine co-transport and was shown to abolish the interaction with V-ATPase-regulator-rag complex [52,68]. In a recent paper by Guo et al. on CTNS protein structure, CTNS^N288K^ was shown to induce structural changes in CTNS favouring a cytosol-open conformation [27]. In addition, CTNS^K280^ was shown to be an important AA in the cystine binding pocket [27,28]. In our study, we showed that CTNS^K280R^ still restored transport activity. As lysine and arginine both have positively charged side chains, this suggests that this mutation has little effect on the binding site, which is consistent with the observation of a juvenile phenotype. The inclusion of CTNS^G339R^ as an additional control validated our model and underscored that not all CTNS^mutants^ rescue cystine accumulation upon stable overexpression: even though the CTNS^G339R^ protein expressed well and located to the lysosomes, in line with other mutants, cystine levels did not lower (Figure 5). We suggest that the substitution of glycine with arginine induces a significant alteration, as it converts a single hydrogen atom into a positively charged side chain. Furthermore, results from a recent study, employing cystinosis mouse PTECs transduced with CTNS^G339R^-HA driven by the strong *CMV* promoter using an adenoviral vector corroborates our findings, as cystine accumulation remained elevated [69]. Therefore, these comprehensive evaluations collectively highlight the G339R mutation’s incapacity to ameliorate cystine accumulation, validating its deleterious effect on CTNS’ function. Despite the recent studies published on CTNS’ structure, it is worth highlighting that our knowledge on the various domains of CTNS, and how mutations influence the domains and CTNS’ function, remains limited [27,28].

The fact that we observed different transport activities than previous studies may be explained by the cell lines employed, together with the overexpression system. Kalatzis and Guo et al. used transient transfection of non-human cells driving *CTNS* cDNA from a potent *CMV* promoter [24,27]. Transient transfection results in acute upregulation of *CTNS* mRNA and protein production, whereas we employed stable transduction using LVs aiming at a single integrated copy and using less potent promoters. Additionally, it is conceivable to consider that the massive overexpression and the relocalisation of CTNS to the plasma membrane in the Kalatzis setup may affect functional aspects important for stabilization, CTNS conformation, and/or for the biological function of CTNS. Lastly, our in vitro data underscore that in silico predictive tools like AlphaMissense and CADD scores are interesting, but that functional assays are still required to corroborate these predictions. Discrepancies in AlphaMissense pathogenicity scores, CADD scores, and our in vitro experiments reveal challenges in correlating the effect of CTNS^mutants^ on protein structure with the degree of cystine transport. This underscores the limitations in the use of predictive tools for cellular and functional phenotypes, even though these tools are instructive in genome-wide association studies to provide an initial judgement for a newly detected mutation.

Our study encountered notable limitations. First, the lack of reliable antibodies to detect CTNS endogenously presented a significant challenge and made us unable to detect endogenous CTNS at the protein level and difficult to interpret protein-to-mRNA data. This limitation necessitated the use of LVs containing 3HA-tagged *CTNS* cDNA. However, it is important to note that our study opted for stable expression over transient approaches, ensuring protein expression levels as closely as possible resembling endogenous expression levels and a CTNS protein locating to the lysosomal membrane.

In our approach, we used a clinically relevant human cell model (*CTNS*^−/−^ CRISPRed ciPTECs), enhancing the translational relevance of our findings, and relied on metabolomic analysis to assess cystine accumulation [32]. Our findings indicate that several CTNS^mutants^ are functional in our addback model (cystine transport), however it is important to mention that the same mutants in patients still result in disease (infantile and juvenile cystinosis), with poor clinical outcome measures, like for kidney survival. This discrepancy may in part be due to the use of cells in the clinical assay that cycle less and accumulate more cystine, whereas our in vitro cell models proliferate and therefore overall carry lower cystine concentrations. Although these limitations are intrinsic to the current state of the field, our methodology aimed to mitigate potential biases and uphold scientific credibility. Taken together, our study underscores that there is a need for a more nuanced interpretation of *CTNS* mutations, revealing variable cystine transport activity for different *CTNS* mutations, similar to observations in cystic-fibrosis-associated *CFTR* mutations [70]. For example, a subset of mutants may result in folding or trafficking errors that in turn influence protein stability, and result in reduced CTNS protein at the lysosomal membrane. Indeed, a drug shown to improve protein folding (chemical chaperones such as CFTR corrector, Corr4a) was shown to restore 70% of the cystine accumulation in patient-derived fibroblasts carrying CTNS^del.67-73^ [49]. Additionally, frameshifts, splicing, or nonsense mutations can result in a premature stop codon leading to little or no CTNS protein expression. Since 15% of patients with cystinosis have nonsense mutations (most common mutation: CTNS^W138X^), there is a possibility to apply translational readthrough [43]. Helip-Wooley et al. showed that gentamycin-induced readthrough of exogenous CTNS^W183X^-GFP in HEK293Ts and in patient-derived cystinosis fibroblasts heterozygous for W138X led to reduced cystine accumulation [71]. Brasell et al. further showed that geneticin (G418) treatment induced translational readthrough of CTNS^W138X^ constructs transfected in HEK293Ts and expression of full length CTNS in homozygous W138X fibroblasts resulting in decreased cystine accumulation [72]. As these compounds are known to cause renal and cochlear toxicity, a modified aminoglycoside without toxicity, called ELX-02, was developed. This aminoglycoside is currently in a phase 2 clinical trial for cystic fibrosis. This novel aminoglycoside produced a functional CTNS protein and reduced cystine accumulation, comparable to cysteamine treatment in cystinosis mice and CTNS^W138X^-cultured fibroblasts without displaying cyto- and nephrotoxicity [73]. Identifying and characterizing these mutations will allow us to create a functional classification of CTNS mutants, as was installed for CFTR. In addition, the identification of specific mutation-induced functional effects provides a foundation for the development of precision medicine for cystinosis as is seen for cystic fibrosis, where depending on the class of functional defect, a different therapy is put forward [70].

The complexity surrounding the correlation between the different clinical phenotypes (infantile, juvenile, and ocular) and the mutations affecting *CTNS* is a challenging issue. Since there is considerable variability in patient phenotype, the use of genotype to make statements of prognosis is not recommended. In a recent international cohort study by Emma et al., with genetic data available for 329 individuals, 33% exhibited homozygosity for the common 57-kb deletion, 23% were heterozygous for the same deletion, and 45% had other pathogenic *CTNS* variants [23]. No apparent differences in kidney survival were observed between patients with homozygous or heterozygous 57-kb deletions and those with other pathogenic *CTNS* variants. Similarly, a study by Veys et al. involving 52 patients from 26 pairs of index and sibling patients, found no significant difference in the age at ESKD between those with homozygous 57 kb deletions and those with other pathogenic variants. However, both studies’ lack of information on the *CTNS* variants’ severity, particularly regarding missense variants, is determined. This underscores the need for more nuanced classification within this category and considering heterozygosity with the 57 kb deletion. An additional layer of complexity is introduced by the various (unknown) functions of CTNS as it is suggested that its role extends beyond mere cystine transport [64]. However, cystine accumulation remains the major hallmark of the disease, and a cornerstone for both diagnosis and treatment. In addition, cystine depletion was shown to be the determining factor, in contrast to the genotype, defining disease outcome (kidney survival) [23]. Most *CTNS* missense mutations have not been functionally characterized, and for most the pathogenic potential remains unclear. To decipher the correlation between *CTNS* mutations and the functional and clinical phenotypes in cystinosis, integrating computational tools with empirical data remains crucial. Moreover, analysing clinical data from well-established cystinosis patient-cohorts focused on specific genotypes is of great interest and is paramount to understand the disease mechanism and to help establish precision medicine. ERKNet and RaDiCo ECYSCO are actively working on a big cohort dataset that includes the patient genotypes and clinical phenotypes for cystinosis [74,75].

In conclusion, we here present an alternative cell model to assess CTNS^mutants^ function to better understand discrepancies between genotype, cystine transport function, and clinical phenotype. Our findings indicate that several CTNS^mutants^ do possess residual transport activity. It should be further explored whether patients carrying missense mutants having some residual transport activity have a milder cystinosis phenotype than those with mutations completely ablating cystine transport. These results indicate that at least for some mutants pharmacologically, an improvement of protein stability in patient-derived cells may result in (partial) rescue of the cystine accumulation phenotype. Additional in vitro studies to examine the effect of CTNS^mutant^ expression on cellular processes are needed to determine the potential of a therapeutic approach and to exclude negative effects.

## Figures and Tables

**Figure 1 cells-13-00646-f001:**
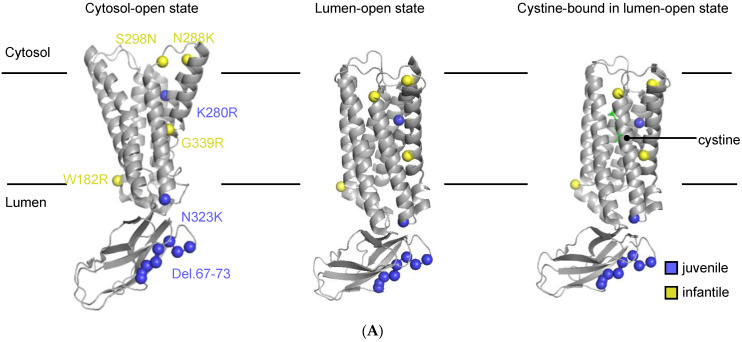
Schematic overview of the CTNS protein with the mutations studied annotated. (**A**) Cryo-EM structures of CTNS in cytosol-open (8DKE), lumen-open (8DKI), and cystine-bound lumen-open (8DKM) state. Mutations are shown in spheres and coloured by clinical phenotypes. Structures indicated were viewed with PyMol 2.4.1. (**B**) Topology of CTNS with mutations indicated and coloured by clinical phenotypes.

**Figure 2 cells-13-00646-f002:**
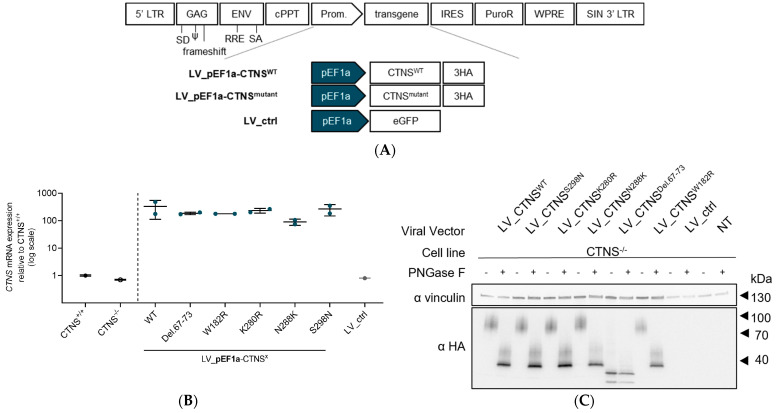
*CTNS^WT^* and *CTNS^mutant^* cDNA addition after lentiviral vector transduction in CRISPRed *CTNS*^−/−^ ciPTECs greatly reduces the intracellular cystine levels. (**A**) Schematic representation of the lentiviral transfer plasmid setup encoding CTNS^WT^, CTNS^mutant^, or eGFP cDNA used to produce the respective lentiviral vectors. *CTNS^−/−^* ciPTECs were transduced with lentiviral vectors expression of CTNS^mutant^-3HA and selected using puromycin to obtain ciPTECs expressing at least 1 integrated copy of the transgene construct. (**B**) *CTNS* mRNA expression level analysis (RT-qPCR) on *CTNS*^−/−^ ciPTECs transduced with lentiviral vectors LV_pEF1a-CTNS^WT^, LV_pEF1a-CTNS^mutant^, or LV_ctrl. The data are normalized for total mRNA levels of γ-Actin and are presented as the mean ± SD (n = 2). (**C**) Western blot analysis of CTNS-3HA protein expression in *CTNS*^−/−^ ciPTECs transduced with lentiviral vectors LV_pEF1a-CTNS^WT^, LV_pEF1a-CTNS^mutant^, or LV_ctrl. Samples were treated with or without PNGase to remove N-glycosylations and normalized for total proteins of vinculin. (**D**) Confocal microscopy images of the immunofluorescence signal of CTNS-3HA (HA, green pseudocolour signal, 633 nm laser) and LAMP1 (LA1, red pseudocolour signal, 561 nm laser) in *CTNS*^−/−^ ciPTECs transduced with either LV_pEF1a-CTNS^WT^, LV_pEF1a-CTNS^mutant^, or LV_ctrl. Nuclei were stained with DAPI. Scale bars are 20 µM. (**E**) Cystine measurement (mass spectrometry) of *CTNS*^−/−^ ciPTECs transduced with either LV_pEF1a-CTNS^WT^, LV_pEF1a-CTNS^mutant^, or LV_ctrl. The data are presented as the mean ± SD (n = 3 or 6 independent metabolite extracts). Statistical testing was performed with a one-way Anova, Sidak’s multiple comparison test. LTR, long terminal repeats; SD, splice donor site; RRE, rev-responsive element; SA, splice acceptor site; cPPT, central polypurine tract; IRES, internal ribosomal entry site; WPRE, woodchuck hepatitis virus posttranscriptional regulatory element; SIN, self-inactivating; LV, lentiviral vector; p, promoter; WT, wild-type; LV_ctrl, LV_pEF1a-eGFP; NT, non-transduced; ****, *p* < 0.0001.

**Figure 3 cells-13-00646-f003:**
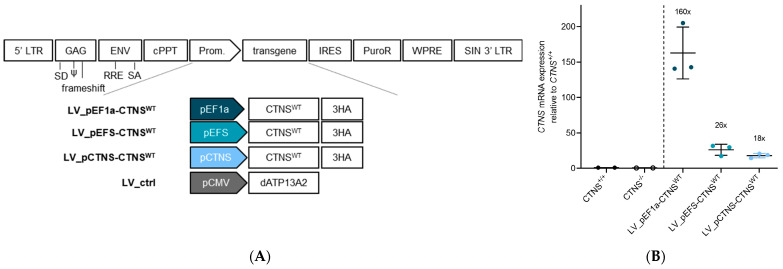
Rescue of the cystinosis phenotype by lower CTNS^WT^ expression enabled by the less potent EFS and CTNS promoters. (**A**) Schematic representation of the lentiviral transfer plasmid setup encoding CTNS^WT^ cDNA driven by either the *EF1a*, *EFS*, or *CTNS* promoter or dATP13A2 cDNA driven by the *CMV* promoter used to produce the respective lentiviral vectors, indicated below. *CTNS^−/−^* ciPTECs were transduced with lentiviral vectors expression of CTNS^WT^-3HA and selected using puromycin to obtain ciPTECs expressing at least 1 integrated copy of the transgene construct. (**B**) *CTNS* mRNA expression level analysis (RT-qPCR) in *CTNS*^−/−^ ciPTECs transduced with lentiviral vectors LV_pEF1a-CTNS^WT^, LV_pEFS-CTNS^WT^, and LV_pCTNS-CTNS^WT^. The data are normalized for total mRNA levels of γ-Actin and are presented as the mean ± SD (n = 3). (**C**) Western blot analysis of CTNS^WT^-3HA protein expression in *CTNS*^−/−^ ciPTECs transduced with lentiviral vectors LV_pEF1a-CTNS^WT^, LV_pEFS-CTNS^WT^, LV_pCTNS-CTNS^WT^, or LV_ctrl. Samples normalized for total proteins of vinculin. (**D**) Cystine measurement (mass spectrometry) of *CTNS*^−/−^ ciPTECs transduced with either LV_pEF1a-CTNS^WT^, LV_pEFS-CTNS^WT^, LV_pCTNS-CTNS^WT^, or LV_ctrl. The data are presented as the mean ± SD (n = 3 or 6 independent metabolite extracts). Cystine (Abundance) was normalized to protein content (µg/µL). Statistical testing was performed with a one-way Anova, Sidak’s multiple comparison test. LV, lentiviral vector; p, promoter; WT, wild-type; LV_ctrl, LV_pCMV-dATP13A2; NT, non-transduced; ****, *p* < 0.0001.

**Figure 4 cells-13-00646-f004:**
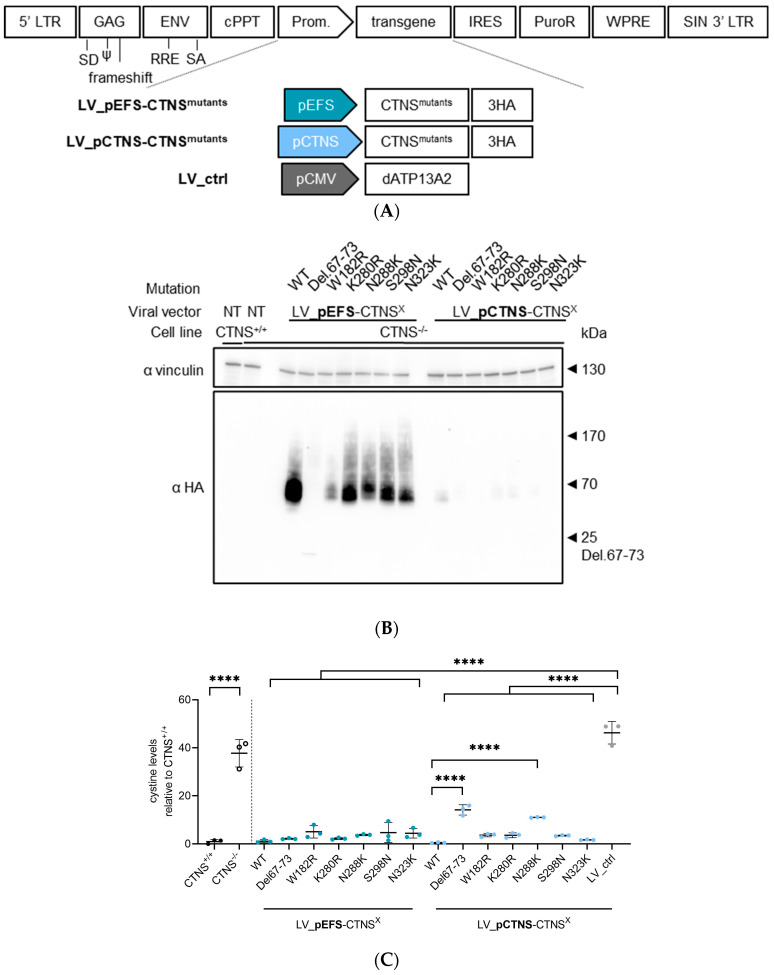
Different promoters exhibit rescue of mutants even with low activity. (**A**) Schematic representation of the lentiviral transfer plasmid setup encoding CTNS^mutants^ cDNA driven by either the *EF1a*, *EFS*, or *CTNS* promoter or dATP13A2 cDNA driven by the *CMV* promoter used to produce the respective lentiviral vectors, indicated below. *CTNS^−/−^* ciPTECs were transduced with lentiviral vectors expression of CTNS^WT^-3HA or CTNS^mutant^-3HA and selected using puromycin to obtain a ciPTECs expressing at least 1 integrated copy of the transgene construct. (**B**) Western blot analysis of CTNS^WT or mutant^-3HA protein expression in *CTNS*^−/−^ ciPTECs transduced with lentiviral vectors LV_pEFS-CTNS^WT or mutant^, LV_pCTNS-CTNS^WT or mutant^, or LV_ctrl. Samples normalized for total proteins of vinculin. (**C**) Cystine measurement (mass spectrometry) of *CTNS*^−/−^ ciPTECs transduced with lentivectors LV_pEFS-CTNS^WT or mutant^, LV_pCTNS-CTNS^WT or mutant^, or LV_ctrl. The data are presented as the mean ± SD (n = 3 independent metabolite extracts). Statistical testing was performed with a one-way Anova, Sidak’s multiple comparison test. LV, lentiviral vector; p, promoter; WT, wild-type; LV_ctrl, LV_pCMV-dATP13A2; NT, non-transduced; ****, *p* < 0.0001.

**Figure 5 cells-13-00646-f005:**
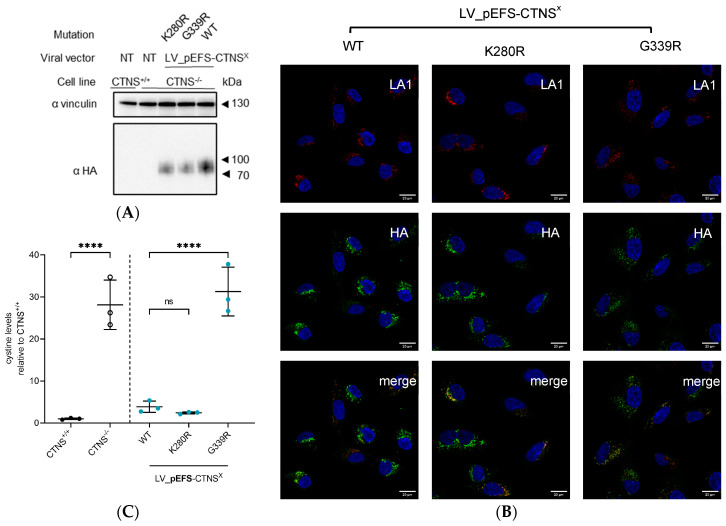
G339R mutant overexpression shows no rescue in cystine accumulation upon cDNA addition in *CTNS*^−/−^ ciPTECs. (**A**) Western blot analysis of CTNS^X^-3HA protein expression in *CTNS*^−/−^ ciPTECs transduced with lentiviral vectors LV_pEFS-CTNS^WT^ or pEFS-CTNS^K280R or G339R^. Samples normalized for total proteins of vinculin. (**B**) Confocal microscopy images of the immunofluorescence signal of CTNS^X^-3HA and LAMP1 in *CTNS*^−/−^ ciPTECs transduced with either LV_pEFS-CTNS^WT^ or LV_pEFS-CTNS^K280R or G339R^. Nuclei were stained with DAPI. Scale bars are 20 µM. (**C**) Cystine measurement (mass spectrometry) of *CTNS*^−/−^ ciPTECs transduced with lentiviral vectors LV_pEFS-CTNS^WT^ or pEFS-CTNS^K280R or G339R^. The data are presented as the mean ± SD (n = 3 independent metabolite extracts). Statistical testing was performed with a one-way Anova, Sidak’s multiple comparison test. LV, lentiviral vector; p, promoter; WT, wild-type; NT, non-transduced; ****, *p* < 0.0001; ns, nonsignificant.

**Table 1 cells-13-00646-t001:** Overview of studied CTNS mutations.

gDNA Mutation	Exon	Protein Mutation	Location	Phenotype	Cystine Transport Activity (%) ^A^	AlphaMissense Score ^B^	CADD Score ^C^	References
c.198_218del21	Exon 5	ITILELP-Del.67–73	N-terminal tail	Juvenile	19 ± 6.1	NA	NA	[24,43,44,45,46,47,48,49]
c.544T > C	Exon 8	W182R	2nd TM	Infantile	34 ± 5	0.7778	26.9	[24,43]
c.839A > G	Exon 10	K280R	5th inter-TM loop	Juvenile	0.68 ± 0.9	0.6602	35	[15,24,27,28,50,51,52]
c.864C > A	Exon 11	N288K	5th inter-TM loop	Infantile	1.6 ± 1.2	0.9942	25	[24,27,45,52,53]
c.893G > A	Exon 11	S298N	5th inter-TM loop	Infantile	77 ± 21	0.5957	29.8	[24,43]
c.969C > G	Exon 11	N323K	6th inter-TM loop	Juvenile	0.14 ± 0.8	0.7791	22.6	[24,45,50,52,54,55]
c.1354G > A	Exon 12	G339R	7th TM	Infantile	−0.8 ± 3.3	0.9803	31	[24,43,56,57,58,59,60,61,62,63]

^A^ Cystine transport activity is expressed as mean percentage of WT CTNS activity ± SEM as determined by Kalatzis et al., 2004 [24]. ^B^ AlphaMissense pathogenicity core: likely benign, 0–0.34; ambiguous, 0.34–0.56; likely pathogenic, 0.564–1.0. ^C^ CADD scores: less likely pathogenic, 1–10; moderate potential to be pathogenic, 10–20; likely pathogenic, 20–99. NA, non-applicable.

## Data Availability

All data are available on request to Rik Gijsbers (rik.gijsbers@kuleuven.be).

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
