# Peer review of "Residual Cystine Transport Activity for Specific Infantile and Juvenile CTNS Mutations in a PTEC-Based Addback Model"

_cells, 2024, doi:10.3390/cells13070646_

Round 1

Reviewer 1 Report

Comments and Suggestions for Authors

The manuscript by Medaer et al. characterizes the cystinosis patient mutants using a clinically relevant cell model (kidney-derived proximal tubule epithelial cell).  In comparison with previous studies expressing CTNS on the plasma membrane for the ease of measuring transport activity, this study took the challenge to express/measure CTNS transport activity in its correct subcellular location, the lysosomes.  The authors also compared different expression levels and surprisingly most of the patient mutation CTNS tested possess transport activity (decrease/rescue the accumulation of cystine in CTNS-/- cells), regardless of the CTNS protein levels (except G339R).  This is in drastic contrast to previous studies, e.g. by Kalatzis et al., showing transport defects in many of the patient CTNS mutations.  These results certainly raise the question as what the best approaches are for studying the function of the CTNS and the patient mutations. 

Some comments

Fig 2D, the resolution of the confocal images is too low to visualize the lysosomal localization and colocalization with lamp1; the control panel (LV_ctrl), the cells were expressing eGFP, so how can they also be stained with HA and visualized with green/laser 488nm showing no signals?

Fig 2C, the molecular weight of the CTNS del67-73 seems to be off (way too small), and no glycosylation occurs (same molecular weight -/+ PNGase F treatment), in contrast to previous study (ref 45) showing DITILELP can be glycosylated).  Did the authors check if this is somehow a truncated form with N-terminal heavy glycosylated region been removed (and therefore no glycosylation in the rest of the protein) ?

For the figures showing cystine levels relative to CTNS+/+, what does the Y-axis represent? is it the fold change? The numbers seem varying quite a bit, e.g. in fig 2E, CTNS-/- level is ~100, while in fig 3D is ~7, in fig4C is ~40, in fig 5C is ~28

In the discussion, the authors mentioned previous studies used transient, acute overexpression v.s. their stable transduction with low integrated copies and less potent promoters.  Potentially the authors’ method did better mimic the patient mutant CTNS (with more relevant cells and localization), yet these patient mutants CTNS managed to reduce/rescue cystine accumulation, which do not behave the same as in patients.  These discrepancies made the reviewer wondering if there are other factors to consider in these studies, e.g. different ways to measure transport activity (whole cell cytosine level with mass spec analysis vs cytosine uptake from outside the cell with isotope cystine measurement).

Typo: in Funding: “his” work…should be “this”

Author Response

We thank the reviewers for their thorough reading, the constructive feedback and suggestions that improved the quality of our revised manuscript.

Below we provide a point-by-point overview for the respective issues raised and how we propose to adapt the manuscript (our replies and suggestions are indicated in blue text). The adapted or new text fragments are indicated with italic text between quotation marks.

In addition, we provide a track changes version of the manuscript and a version with the track changes accepted.

Authors reply to Reviewer 1

Reviewer 1: The manuscript by Medaer et al. characterizes the cystinosis patient mutants using a clinically relevant cell model (kidney-derived proximal tubule epithelial cell).  In comparison with previous studies expressing CTNS on the plasma membrane for the ease of measuring transport activity, this study took the challenge to express/measure CTNS transport activity in its correct subcellular location, the lysosomes.  The authors also compared different expression levels and surprisingly most of the patient mutation CTNS tested possess transport activity (decrease/rescue the accumulation of cystine in CTNS-/- cells), regardless of the CTNS protein levels (except G339R).  This is in drastic contrast to previous studies, e.g. by Kalatzis et al., showing transport defects in many of the patient CTNS mutations.  These results certainly raise the question as what the best approaches are for studying the function of the CTNS and the patient mutations. 

Some comments:

Fig 2D, the resolution of the confocal images is too low to visualize the lysosomal localization and colocalization with lamp1; the control panel (LV_ctrl), the cells were expressing eGFP, so how can they also be stained with HA and visualized with green/laser 488nm showing no signals?

We apologize for the low resolution of the respective images in the figure. We believe this is caused by the conversion to pdf while uploading the manuscript in the portal. High resolution images (330 dpi) were added to the manuscript to enhance the visual appreciation of co-localization of CTNS-3HA with LA1.

The primary antibodies (Ab) to detect CTNS-3HA and LAMP1 were the following: mouse monoclonal Ab (mAb) HA.11 (BioLegend 901515) and rabbit mAb D2D11 (anti-LAMP1 - Cell Signaling 9091), respectively. As secondary Abs, goat anti-mouse Alexa 633 (Life Technologies A21050), and goat anti-rabbit Alexa 555 (Invitrogen A21429) were used. The lasers used to visualize the HA- and LA1-signals were 633 and 561 nm, respectively. After imaging, the magenta pseudocolour was changed to green in the ZEN 3.4 software to enhance the visibility in the merge-image. These lasers were opted to avoid excitation of the eGFP fluorophore and to accurately detect background fluorescence in the LV_ctrl condition for HA (633 nm). These adjustments were made to ensure clearer and a more accurate representation of the data.

The legend of Figure 2D and Supplemental Table S2 were adapted to provide clarification regarding the confocal microscopy set-up:(D) Confocal microscopy images of the immunofluorescence signal of CTNS-3HA (green pseudocolour, 633 nm laser) and LAMP1 (red pseudocolour, 561 nm laser) in CTNS-/- ciPTECs transduced with either LV_pEF1a-CTNSWT, LV_pEF1a-CTNSmutant, or LV_ctrl.”

Fig 2C, the molecular weight of the CTNS del67-73 seems to be off (way too small), and no glycosylation occurs (same molecular weight -/+ PNGase F treatment), in contrast to previous study (ref 45) showing DITILELP can be glycosylated).  Did the authors check if this is somehow a truncated form with N-terminal heavy glycosylated region been removed (and therefore no glycosylation in the rest of the protein) ?

Thank you for brining up the discrepancy in the molecular weight of CTNSdel.67-73 observed in Figure 2C. Nevo et al (2017) showed a molecular weight of around 60 kDa. In this study, CTNSdel67-73 is actually CTNS tagged with an eGFP (around 27 kDa) explaining the higher molecular weight compared to our 3HA-tagged (only 3 kDa) CTNSde67-73 (around 35 kDa). A more recent study by Venkatarangan et al. (2023) showed CTNSdel67-73 exists in different forms: partially glycosylated full-length, protein product translated from internal methionine, non-glycosylated full-length protein, or N-terminal truncated product. In line with our results, deglycosylation by PNGase F, removing all N-linked oligosaccharides, did not alter the molecular weight of CTNSdel67-73, which underscores the presence of non-glycosylated full length CTNSdel67-76. In the same study they also showed a smaller band resulting in an N-truncated product, which we also observe.

For the figures showing cystine levels relative to CTNS+/+, what does the Y-axis represent? is it the fold change? The numbers seem varying quite a bit, e.g. in fig 2E, CTNS-/- level is ~100, while in fig 3D is ~7, in fig4C is ~40, in fig 5C is ~28.

The reviewer is correct. The cystine levels are shown as fold change relative to CTNS+/+ (cystine nmol/mg protein). We opted for this representation to acknowledge the importance of a meaningful comparison between the different conditions. Technical limitations at the metabolomics core facility necessitated us to use a different quantification in Figure 3D. In this experiment absolute quantification was not possible due to difficulties with the internal standard and thus we only obtained abundancies in this run, which still allows comparison between conditions. Due to these constraints, cystine levels were represented as cystine abundance (cystine abundance/µg protein) instead of cystine concentration (cystine nmol/mg protein). Differences were expressed as fold-change compared to the wild-type CTNS+/+ samples. However, the low cystine baseline levels in CTNS+/+ samples do still vary between experiments (even though we measured triplicate for each condition), and lead to significant fold change differences even with small deviations. For example, in Figure 2E, 4C, 5C, the cystine levels for CTNS+/+ were very low with values of 0.01, 0.05, and 0.12 nmol/mg protein, respectively. Hence, this variability in the baseline levels of CTNS+/+ can impact the interpretation of fold change differences. We provided the values therefore as well in the Supported Data Values file that was submitted with the manuscript.

In the discussion, the authors mentioned previous studies used transient, acute overexpression v.s. their stable transduction with low integrated copies and less potent promoters.  Potentially the authors’ method did better mimic the patient mutant CTNS (with more relevant cells and localization), yet these patient mutants CTNS managed to reduce/rescue cystine accumulation, which do not behave the same as in patients.  These discrepancies made the reviewer wondering if there are other factors to consider in these studies, e.g. different ways to measure transport activity (whole cell cytosine level with mass spec analysis vs cytosine uptake from outside the cell with isotope cystine measurement).

We thank the reviewer for this comment and would like to add some clarification. In patient samples cystine is determined in white blood cells (WBC), whereas we determined cystine levels in human PTECs (Wilmer et al. 2011 Pediatr Nephrol). In line, our study used whole cell cystine measurements via mass spectrometry to estimate the transport activity of CTNS, thus providing us with physiologically more relevant results.

In addition, compared to prior studies, we here opted for a more physiological relevant set-up by expressing CTNS WT and CTNS mutants at the lysosomal membrane at lower expression levels. Even though we aimed to mimic the situation as closely as possible as in people living with cystinosis, the PTEC model still is an immortalized cell model, which implies that cells divide and thus cystine concentration may be underestimated.

We do acknowledge that patients with the respective mutations do show accumulation in their WBCs. In our study several missense CTNS mutants rescued cystine levels, whereas the G339R mutant did not rescue cystine accumulation, validating the model in our opinion. Importantly, our results demonstrate that at least a subset of CTNS mutants still allows transport of cystine, contrary to what was previously understood. The fact that these CTNS mutants in our model rescue cystine accumulation may be due to different reasons: (i) we employ dividing PTECs instead of primary WBCs (ii) the CTNS expression levels we obtain are still higher compared to endogenous CTNS levels, (iii) the current viral vector design contains an mRNA stabilizing element (WPRE) that is also included in the CTNS mRNA driven from the pCTNS promoter; (iv) the current PTEC cells contained 2 integrated copies; (v) our LV vector only contains the open reading frame for CTNS, and lacks the untranslated and the intronic regions, that are known to be important in RNA stability; (vi) the CTNS-promoter used only contains 800 bp of the CTNS promoter sequences, and may be more potent than the endogenous promoter.

Typo: in Funding: “his” work…should be “this”

We thank the reviewer for pointing out this typo. The typo was corrected in the manuscript: “Funding: This work was funded by FWO PhD scholarship of D. David (180936-1S22921N-SW), FWO grant (G056521N) to Rik Gijsbers, and the KU Leuven C1 Grant to Elena Levtchenko and Rik Gijsbers (C14/17/11).”

Reviewer 2 Report

Comments and Suggestions for Authors

Medaer et al investigated the phenotype correlations, subcellular localisation and function of clinically relevant CTNS mutants with CTNS depleted proximal tubule epithelial cells and patient-derived fibroblasts. They found that some mutants could still reverse cystine accumulation, but G339R mutant did not. The authors thus advocate robust assessment methodologies and urge more clinically relevant models in studying cystinosis. Overall, the manuscript is well-written and easy to read. However, I do have some concerns, listed below:

1. The authors using qPCR quantification to measure CTNS expression, but didn’t use western blot, which really limited this study. Because it is hard to know the real expression of CTNS with only qPCR, and thus hard to interpret the correlation of CTNS WT and mutants expression and the phenotypical data.  

2. Also because of the above reason, they should consider doing amplicon sequencing to confirm efficient CRISPR gene editing in the CTNS-/- cell line.   

3. Regarding the CTNS-/- cell line, how was this cell line established? Did it involve permanent expression of Cas9 and sgRNA? I would assure it was a transient expression to KO CTNS, otherwise this will interfere thereafter CTNSmutant expression.

4. The authors tried different promoters to drive CTNS wildtype and mutants expression, and they found that EF1a promoter was too strong and resulted in supraphysiological expression, then they tried EFS promoter and CTNS promoter, and they got much lower expression, much closer to the endogenous expression level. However, even the lowest expression (with CTNS promoter) was still 18x higher on mRNA level, this make it really hard to justify that the phenotype of lower cystine accumulation is driven by physiological expression of the CTNS mutant.    

5. Figure 2C, - or + signs of PNGase F are not aligned well with the corresponding western blot, make it hard to interpret the results, please modify it.

Author Response

We thank the reviewers for their thorough reading, the constructive feedback and suggestions that improved the quality of our revised manuscript.

Below we provide a point-by-point overview for the respective issues raised and how we propose to adapt the manuscript (our replies and suggestions are indicated in blue text). The adapted or new text fragments are indicated with italic text between quotation marks.

In addition, we provide a track changes version of the manuscript and a version with the track changes accepted.

Authors reply to Reviewer 2

Reviewer 2: Medaer et al investigated the phenotype correlations, subcellular localisation and function of clinically relevant CTNS mutants with CTNS depleted proximal tubule epithelial cells and patient-derived fibroblasts. They found that some mutants could still reverse cystine accumulation, but G339R mutant did not. The authors thus advocate robust assessment methodologies and urge more clinically relevant models in studying cystinosis. Overall, the manuscript is well-written and easy to read. However, I do have some concerns, listed below:

1. The authors using qPCR quantification to measure CTNS expression, but didn’t use western blot, which really limited this study. Because it is hard to know the real expression of CTNS with only qPCR, and thus hard to interpret the correlation of CTNS WT and mutants expression and the phenotypical data.  

We apologize for any confusion regarding the quantification of CTNS expression. We did rely on Western blot analysis to compare CTNS-3HA tagged protein expression levels for the different promoters. However, endogenous CTNS cannot be detected using antibodies (in our hands we were not able to validate any of the commercially available antibodies for CTNS) and cannot be visualized using an HA-tag antibody. Therefore we relied on qPCR to assess relative expression levels of our stable cell lines compared to the endogenous CTNS.

We would like to clarify that the manuscript contains western blot analysis to compare the CTNS-3HA protein expression of the different mutants with CTNS-3HA WT: Figure 2C, Figure 3C, Figure 4B, and Figure 5A. Quantification of these western blots can be found in the supplemental figures: Figure S3C, figure S4C. We totally agree that western blot analysis is crucial for assessing protein expression levels, and we have provided the necessary data to support our findings.

2. Also because of the above reason, they should consider doing amplicon sequencing to confirm efficient CRISPR gene editing in the CTNS-/- cell line.   

3. Regarding the CTNS-/- cell line, how was this cell line established? Did it involve permanent expression of Cas9 and sgRNA? I would assure it was a transient expression to KO CTNS, otherwise this will interfere thereafter CTNSmutant expression.

The CTNS-/- KO ciPTEC line was generated by the group of prof. Masereeuw (Utrecht university). This work has previously been published: Jamalpoor, A.; Gelder, C.A.G.H. Van; Yousef Yengej, F.A.; Zaal, E.A.; Berlingerio, S.P.; Veys, K.R.; Pou Casellas, C.; Voskuil, K.; Essa, K.; Ammerlaan, C.M.E.; et al. Cysteamine–Bicalutamide Combination Therapy Corrects Proximal Tubule Phenotype in Cystinosis. EMBO Mol. Med. 2021, 13, e13067, doi:10.15252/emmm.202013067. Details on the generation of the CTNS KO clone are provided in this paper. CTNS was knocked out using CRISPR/Cas9 technology with a guide RNA targeting exon 4 of CTNS by transfection. After cell sorting, clonal cell expansion was done. Knock-out was confirmed by Sanger sequencing showing 13 and 85 bp deletion, respectively, followed by intracellular cystine measurement and accumulation of cystine (5.19± 0.30 versus 0.05 ± 0.02 nmol/mg for WT control ciPTECs). We have indicated this in the materials and method section.

4. The authors tried different promoters to drive CTNS wildtype and mutants expression, and they found that EF1a promoter was too strong and resulted in supraphysiological expression, then they tried EFS promoter and CTNS promoter, and they got much lower expression, much closer to the endogenous expression level. However, even the lowest expression (with CTNS promoter) was still 18x higher on mRNA level, this make it really hard to justify that the phenotype of lower cystine accumulation is driven by physiological expression of the CTNS mutant.    

We acknowledge the concern of Reviewer 2, and have taken the opportunity to address it in our discussion (line nr 477). We agree that the expression currently is still higher than endogenous expression, but compared to other studies, we are convinced expression levels in our experiments are physiologically more relevant. Previous studies resulted in higher expression levels of CTNS by employing the CMV-promoter (Kalatzis et al 2004; Guo et al 2022). In contrast, our approach employed different promoter constructs, among which also the CTNS-promoter driven constructs, which demonstrated expression levels closer to physiological levels. Western blot analyses revealed that CTNS expression substantially decreased with promoter strength, reaching levels that are close to the detection limit for CTNS-promoter driven CTNS, indicating a more controlled expression profile. However, it is to note that lower expression levels pose limitations on the techniques used for analysis (for example, it will be impossible to detect a significant protein signal).

As indicated in the reply to reviewer 1, the fact that these CTNS mutants in our model express CTNS at higher levels than endogenous CTNS may be due to different reasons: (i) we employ dividing PTECs instead of primary WBCs, (ii) the CTNS expression levels we obtain are still higher compared to endogenous CTNS expression levels, (iii) the current viral vector design contains an mRNA stabilizing element (WPRE) that is also included in the CTNS mRNA driven from the CTNS-promoter; (iv) the current PTEC cells contain 2 integrated copies; (v) our LV vector only contains the open reading frame for CTNS, and lacks the untranslated and the intronic regions, that are known to be important in RNA stability; (vi) the CTNS-promoter used only contains 800 bp of the CTNS-promoter sequences, and may be more potent than the endogenous promoter.

5. Figure 2C, - or + signs of PNGase F are not aligned well with the corresponding western blot, make it hard to interpret the results, please modify it.

We thank the reviewer for pointing out the misalignment of the symbols. Our manuscript was adapted to enhance the readability of the western blot in Figure 2C.

Round 2

Reviewer 2 Report

Comments and Suggestions for Authors

The authors addressed all my concerns.